# Growth Monitoring and Yield Estimation of Maize Plant Using Unmanned Aerial Vehicle (UAV) in a Hilly Region

**DOI:** 10.3390/s23125432

**Published:** 2023-06-08

**Authors:** Sujan Sapkota, Dev Raj Paudyal

**Affiliations:** 1Faculty of Science, Health and Technology, Nepal Open University, Manbhawan, Lalitpur, Nepal; 2School of Surveying and Built Environment, University of Southern Queensland, Springfield, QLD 4300, Australia

**Keywords:** differential global positioning system (DGPS), precision agriculture, digital surface model (DSM), digital terrain model (DTM), green–red vegetation index, leaf area index (LAI), near infrared (NIR), NDVI, receiver independent exchange format (RINEX)

## Abstract

More than 66% of the Nepalese population has been actively dependent on agriculture for their day-to-day living. Maize is the largest cereal crop in Nepal, both in terms of production and cultivated area in the hilly and mountainous regions of Nepal. The traditional ground-based method for growth monitoring and yield estimation of maize plant is time consuming, especially when measuring large areas, and may not provide a comprehensive view of the entire crop. Estimation of yield can be performed using remote sensing technology such as Unmanned Aerial Vehicles (UAVs), which is a rapid method for large area examination, providing detailed data on plant growth and yield estimation. This research paper aims to explore the capability of UAVs for plant growth monitoring and yield estimation in mountainous terrain. A multi-rotor UAV with a multi-spectral camera was used to obtain canopy spectral information of maize in five different stages of the maize plant life cycle. The images taken from the UAV were processed to obtain the result of the orthomosaic and the Digital Surface Model (DSM). The crop yield was estimated using different parameters such as Plant Height, Vegetation Indices, and biomass. A relationship was established in each sub-plot which was further used to calculate the yield of an individual plot. The estimated yield obtained from the model was validated against the ground-measured yield through statistical tests. A comparison of the Normalized Difference Vegetation Index (NDVI) and the Green–Red Vegetation Index (GRVI) indicators of a Sentinel image was performed. GRVI was found to be the most important parameter and NDVI was found to be the least important parameter for yield determination besides their spatial resolution in a hilly region.

## 1. Introduction

The population of the world has been rising day by day, thus increasing the demand for food, shelter, and other basic needs [1]. Land is the most common natural resource which fulfills all the basic needs providing a platform for food production, shelter, and other basic needs [2]. Land may be thus taken as a finite resource in the sense that its area cannot be increased. Thus, for the increasing population, the only way to maintain food resources is by increasing productivity [3]. Due to the advancement of technology, the use of fertilizers, and other means, productivity can be increased, thus maintaining a balance between population and food resources [4,5]. For viable agricultural production, the study of the latest trends and technology in an agricultural domain is necessary. Tracking the phases of a crop can be achieved by studying its phenology and biomass estimation, which ultimately helps in understanding environmental factors that affect the crop growth and yield it provides [6].

In agricultural streams such as forestry and crop production, biomass is normally defined as the dry mass of the above-ground part of a specific category of plants [7,8]. Biomass is important in various fields as it provides much information regarding plant growth, its yield, energy that can be liberated from it, and so on [9]. Therefore, examining biomass is helpful for many research and forecast activities [10]. Biomass can be examined with various methods, i.e., direct burning and weighing, and using empirical formulas for specific plants.

Remote sensing products such as Vegetation Indices are often used to estimate biomass and monitor plant growth [10]. Till now, various Vegetation Indices have been developed to monitor plant growth and estimate biomass. Some of them are Normalized Difference Vegetation Index, Soil Adjusted Vegetation Index, Green Vegetation Index, Green–Red Vegetation Index, Excess Green Vegetation Index, etc. [11]. Among them, NDVI and SAVI are considered to be the more common and accurate means to estimate biomass and monitor plant growth [11,12]. However, the calculation of NDVI and SAVI requires an NIR camera; the images seem to be expensive to produce [13]. On the other hand, ExG and GRVI provide a means of plant growth and monitoring using RGB cameras and images since they can be easily calculated from images captured from RGB cameras [14,15].

These days, most precision farming investigations are focused on the execution of a wide extent of sensors and instruments able to remotely identify crop and soil properties in quasi-real time [16,17]. The spatial resolution of major satellite sensors has been upgraded dramatically in the modern days [18,19]. However, they are not able to perform repeated measurements regarding the crop cycle. In order to alleviate such problems, the use of Unmanned Aerial Vehicle ((UAV) that allows very high spatial resolution (of the order of a few centimeters), as well as the ability to obtain repeated measurements from time to time, is the advantage over high-altitude remote sensing [20,21]. Within the last decade, the improvement of Unmanned Aerial Vehicle (UAV) platforms characterized by small size has advertised an unused solution for crop management and observing, capable of convenient provision of high-resolution images, particularly where small productive ranges have to be checked [22,23].

This research study aims to capture aerial imagery through UAV and further process those images from which Crop Surface Model (CSM), Plant Height (PH), Green–Red Vegetation Index (GRVI), Biomass, and, finally, Yield values can be obtained using a field-based technique, aerial survey and satellite-based technology [4,24,25]. The result obtained from various sources has been compared with the ground-based actual result to see the deviation from actual yield in the ground [26]. This research project has the potential to revolutionize the way that maize is grown and harvested. By providing farmers with accurate and timely information about plant growth, the project can help them to make better decisions about irrigation, fertilization, and other management practices. This can lead to increased yields and improved profitability for farmers [27,28].

## 2. Materials and Methods

First and foremost, visiting the proposed project site was one preliminary task. GCPs were established and the coordinates of the GCPs were determined with the help of DGPS. Images of the plot only (without crop) were taken with the help of an Unmanned Aerial Vehicle (RGB spectral bands). Then, UAV images were taken at numerous phenological stages of the maize plant life cycle to monitor the crop growth and to estimate yield [29,30]. From the acquired images, Orthophoto, DTM, and DSM were created. DSM is needed for average crop height. From the generated Orthophoto, Green–Red Vegetation Index was calculated. Leaf Area Index (LAI) was calculated from the field-based method in which the area of the leaf is computed with the help of a measuring tape. Around 25 sample points of areas of 1 m^2^ (1 m × 1 m) were chosen in the field. The average Plant Height, Biomass, and LAI of the sample points were computed to generalize our result. From the computed values, GRVI, LAI, Biomass and Plant Height, Yield of the crop plot was estimated. Variations in LAI wrt Yield and Biomass, Plant Height wrt Yield and Biomass, GVI wrt Biomass and Yield, GRVI wrt Biomass and Yield and GVI and GRVI wrt Biomass and Yield were modelled with the help of respective graphs between them. NDVI was also calculated using a Sentinel-based product from Google Earth Engine [31]. A Sentinel satellite dataset is a collection of data collected by the European Space Agency’s (ESA) Sentinel satellites. These satellites are part of the Copernicus program, a large Earth observation program which collects data about land, marine, and atmospheric environments. The data collected by the Sentinel satellites are used to monitor and study climate change, natural disasters, land use, and other environmental conditions. Sentinel datasets can be used for a variety of applications, including monitoring of agricultural land, mapping glaciers, assessing deforestation, detecting oil spills, and studying ocean currents. The NDVI product was also used to estimate the yield. Finally, all the parameters were used to determine the yield and validated with the actual yield from the ground.

Figure 1 illustrates the research design which has been implemented for the completion of this study. Both primary and secondary data were used to complete this work. The data collection method is termed a mixed method because both primary and secondary datasets have been used here. Primary data were collected from a field-based survey and remote sensing imagery was the secondary source data. As a part of primary data, the imagery was obtained from the Unmanned Aerial Vehicle. Secondary data source incorporated the use of sentinel based satellite imageries.

Study Area

The study area of the project “Growth Monitoring and Yield Estimation of Maize Plant using UAV” is Dhulikhel, Kavrepalanchowk, as illustrated in Figure 2 below. Dhulikhel is one of the leading municipalities in Kavrepalanchok District of Nepal. Dhulikhel is located at 27°37′20″ North latitude and 85°33′34″ East longitude [32,33]. It is situated at the Eastern edge of Kathmandu Valley, south of the Himalayas at 1550 m over sea level, and lies 30 km southeast of Kathmandu and 74 km southwest of Kodari. B.P Highway and Araniko Highway pass through Dhulikhel, which are the vital highways of Nepal [34,35]. The majority of population is engaged in agriculture; rice, maize and wheat are the major crops of Dhulikhel Municipality [36]. The production of maize seems to be increasing rapidly every year, whereas the production of wheat seems to be decreasing. Winter is characterized by much less rainfall than summer [37,38]. Production of the maize crop is very favorable in this climate, and since the production is growing every year, the study of different phonological stages of maize is needed as has been performed here [39].

Research Design

The overall methodology of the experiment is presented in Figure 3.

## 3. Results

The whole project area was divided into five different sample areas based on the area of the individual plot (Figure 4). Further, the sample area was divided into five different sub-plots based on the clustered sampling technique as 1a, 1b, 1c, 1d and 1e for Sample Area 1. Similarly, for Sample Area 2, these sub-plots were divided into 2a, 2b, 2c, 2d and 2e sample areas. Further, other sub-plots were classified accordingly.

### 3.1. DGPS Survey Result

DGPS survey was carried out in order to establish the Ground Control Points (GCPs). Thus established GCPs were further used for referencing in the images (Figure 5). The base used was the fourth-order control point from the Land Management Training Center.

Obtained coordinates of the respective plots are tabulated below in Table 1.

### 3.2. Growth Monitoring through Leaf Area Index

Leaf Area Index is another approach to monitoring the growth of the plant. The leaf area was measured directly in the field in different growth stages of the plant. The change in Leaf Area Index shows the growth pattern of the plant. LAI was computed using the formula given below:(1)$$\mathrm{LAI}\;=\frac{\;\mathrm{Leaf}\;\mathrm{Area}}{\mathrm{Ground}\;\mathrm{area}}.$$

Figure 6 shows the Leaf Area Index of the maize plant in the project area at different phenological stages (26 days for first phase, 22 days for second phase, 25 days for third phase). According to the graph, LAI value ranges from 0 to 9.55. According to the data obtained on 8 August, the value of the growth of a plant is higher in Sub-plot 5, i.e., 5a, 5b, 5c, 5d and 5e. The LAI on 8 August was affected by rain and wind, causing the leaf of Sub-plot 2e to be zero, meaning that the plant was dead. LAI seems to be continuously increasing in different phenological stages, starting on 25 May 2021, 22 June 2021, 14 July 2021, and ending finally on 15 August 2021.

### 3.3. Growth Monitoring through Crop Surface Model

Crop Surface Model is one of the approaches to observing the development of a plant. The CSM generated at numerous phases helps in monitoring the growth in individual plots. The variation in the plant height at different stages shows the growth pattern of the plant. Generation of the Crop Surface Model from the obtained Digital Surface Model (DSM) was obtained at different phases of crop life cycle as indicated in the figure above.

Figure 7 shows the Crop Surface Models of the project area at different phenological stages. The CSM on 25 May shows that the plant height is in the range of 0 m to 2.4 m. The growth of a plant is maximum in Plot T9 and Plot T10 and low in Plots T3, T4, T5, and T6. The CSM on 23 July shows that the height of plants is in the range of 0 m to 4 m. The growth of a plant is maximum in Plots T9 and T10. The CSM maps show the plot-wise comparison of the plant height at different growth stages. The change in plant height in each plot is seen on the CSM maps generated on a different date. From the above maps, it is seen that the overall growth of a plant is maximum in Plot T9 and Plot T10.

Figure 8 shows the plot-wise plant height of the plot at different growth stages generated from the Crop Surface Model. The variation in plant height in various plots at numerous developmental phases helps in monitoring the growth of a plant. The graph shows that the overall growth of a plant is maximum in Plot 4 and Plot 5. Since Plots 1, 2 and 3 were adversely affected, the Plant Height value in these Plots also seems to be highly affected. In Plot 2e, Plant Height seems to be 0, meaning that all the plants in that sample plot were dead during the image acquisition at the last time frame, i.e., 15 August 2021.

### 3.4. Growth Monitoring through Green–Red Vegetation Index

Green–Red Vegetation Index is another approach to monitor plant growth. The images taken at a different time are processed to obtain an orthomosaic of the images from which the Green–Red Vegetation Index is generated using the ArcGIS 10.8 software.
(2)$$\mathrm{GRVI}\;=\frac{\;\mathrm{Green}-\mathrm{Red}}{\mathrm{Green}+\mathrm{Red}}.$$

Figure 9 shows the Green-Red Vegetation Index at different growth stages. The change in GRVI value shows the growth pattern of the plants. The GRVI map generated on June 22 shows that the GRVI value at that stage ranges from −0.2 to 0.2 and there is a similar pattern of GRVI in each sub-plot, which shows that there is a similar growth pattern of plants in each sub-plot. The negative GRVI value represents that there is the less green pigment in the plant, i.e., the reflectance value of visible red light is greater than the that of visible green light. The GRVI map generated on May 25 shows that the GRVI value at that stage ranges from −0.2 to 0.4 and there are maximum GRVI values in Plots 1 and 3. Similarly, the GRVI map generated on July 14 shows that the GRVI value at that specific time period ranges from 0.2 to 0.4 at different sub-plots. Finally, after the image acquisition on 15 August, GRVI map was generated on which GRVI value slightly declined to that of 14 July. Even at the final growth monitoring stages, GRVI value at different sub-plots declined randomly; this is because the plant in that region was dead because of heavy rain and wind at that time.

Figure 10 shows the plot-wise average GRVI value in each growth stage. The graph shows the change in GRVI value in each growth stage. The GRVI value is higher in Plot 4 and Plot 5, which shows that the value of the growth of a plant is higher in Plot 4 and Plot 5. In some plots, the GRVI value on August 15 decreased from the previous stage, which means that the plant started to become yellowish, i.e., the maturity stage of the plant started.

### 3.5. Relation between Crop Canopy Parameters

#### 3.5.1. Relation between Plant Height and Leaf Area Index

Figure 11 shows the relation between the Plant Height and Leaf Area Index of the plot. The Plant Height and Leaf Area Index of the plants at different growth stages are plotted on a graph to determine the relationship between Plant Height and Leaf Area Index. The graph shows the linear positive relationship between the Leaf Area Index and Plant Height. The regression equation y = 2.3101x + 2.2867 is developed from the graph, where y is the Leaf Area Index, i.e., the dependent variable of the relation, and x is the Plant Height, i.e., the independent variable of the relation. The coefficient of determination of the relation is 0.7992, which shows a very strong relationship between Plant Height and Leaf Area Index. With the increase in Plant Height, the Leaf Area Index also increases. This relation shows that there is a strong relation between Leaf Area Index and Plant Height, therefore Leaf Area Index can be used to monitor the growth of the plant.

Prior to these, some of the outliers can even be seen in the graph, meaning that the final stage of the maize plant was monitored with the acquisition of image on 15 August, and several plants were found dead, so their Leaf Area Index was mismatched.

#### 3.5.2. Relation between GRVI and NDVI

The below graph shows the relation between GRVI and NDVI of the plot (Figure 12). The regression equation between the GRVI and NDVI is y = 1.3785x +0.5373 with a coefficient of determination of 0.5. This shows the mild relationship between GRVI and NDVI. NDVI was generated using a Sentinel-based product in Google Earth Engine. Since the NDVI does not seem to have a strong relation with the GRVI, we use GRVI to estimate the yield of the crop and test whether or not the Sentinel-based product is capable of estimating the yield.

### 3.6. Model Generation, Estimation and Validation

As illustrated in the methodology figure, the overall methodology of the thesis work is clearly shown. Yield estimation and validation were part of the workflow diagram. Estimation of yield was performed as part of a model generation, where the model was generated using regression analysis, which is shown below [40]. Further, the methodology for the validation of the obtained yield is also illustrated in Figure 13 below.

#### 3.6.1. Relation between Yield and Plant Height

Plant Height was generated from the Crop Surface Model, which was used to develop a relationship between Plant Height and Yield. The yield from all the samples measured in the field was used to generate the regression model presented in Table 2.

#### 3.6.2. Relation between Yield and Leaf Area Index

Leaf Area Index was measured from the field measurement where the leaf of every plant of the sample area was measured. T2, T3, T4, and T5 time periods were used to measure the Leaf Area Index and were later used to develop a relationship with Yield. The yield from all samples measured in the field was used to generate the regression model (Table 3).

#### 3.6.3. Relation between Yield and Green–Red Vegetation Index

Green–Red Vegetation Index was calculated in the different time period within the sample plot to see its relation with Yield. T2, T3, T4, and T5 time periods were used to measure the Green–Red Vegetation Index and were later used to develop a relationship with Yield (Table 4).

#### 3.6.4. Relation between Yield and Biomass

Biomass was calculated in the final time period within the sample plot to see its relation with Yield (Table 5).

#### 3.6.5. Relation between Yield and NDVI

Normalized Difference Vegetation Index (NDVI) generated from Sentinel-based products was used to see the relation between NDVI and Yield (Table 6).

#### 3.6.6. Relation between Yield and Satellite-Based GRVI

Green–Red Vegetation Index generated from the Sentinel-based product was used to see the relation between GRVI and Yield (Table 7).

#### 3.6.7. Estimation of Yield from Plant Height

The regression model developed between Yield and Plant Height for the plot was determined using the regression model, which was used to estimate the yield as tabulated below (Table 8).

Yield was estimated from each sample plot using the regression equation [41]. Since 70% of the total data was used for constructing the regression equation, the remaining 30% of the data was used to validate the result [42,43]. The error in percentage for the first sample plot was found to be 21.81%. Similarly, for the second sample plot, an error was computed to be 66.27%, which is the maximum error within all the sample plots. The main reason behind this large error is that the second plot was heavily affected by wind and rain, causing the plant to die. Since the model was generated using the condition that any other environment circumstance hasn’t affected the growth of maize plant, this led to a difference between the actual ground scenario and the model equations, resulting in high error. Moreover, for the third sample plot, the error was found to be 19.64%. The fourth and fifth sample plots contained a lesser amount of error compared to other sample plots as they were the least affected by environmental circumstances, and the error was found to be 7.32% and 3.33%, respectively.

#### 3.6.8. Estimation of Yield from Leaf Area Index (LAI)

The regression model developed between the Yield and Leaf Area Index for the plot was determined using the regression model, which was used to estimate the yield as tabulated below (Table 9).

Yield was estimated from each sample plot using the regression equation [41]. Since 70% of the total data was used for constructing the regression equation, the remaining 30% of the data was used to validate the result. The error in percentage for the first sample plot was found to be 14.15%. Similarly, for the second sample plot, the error was computed to be 58.49%, which is the maximum error within all the plots i.e., 1,2,3,4 and 5. The main reason behind this large error is that the second plot was heavily affected by wind and rain, causing the plant to die. Since the model was generated using the condition that any other environment circumstance has not affected the growth of maize plant, this led to a difference between actual ground scenario and the model equations, resulting in high error. Moreover, for the third sample plot, the error was found to be 15.60%. The fourth and fifth sample plots contained a lesser amount of error compared to other sample plots as they were the least affected by environmental circumstances, and the error was found to be 5.30% and 1.08%, respectively.

#### 3.6.9. Estimation of Yield from Green–Red Vegetation Index (GRVI)

The regression model developed between Yield and Green Red Vegetation Index (GRVI) for the plot was determined using a regression model, which was used to estimate the yield as tabulated below (Table 10).

Yield was estimated from each sample plot using the regression equation [41]. Since 70% of the total data was used for constructing the regression equation, the remaining 30% of the data was used to validate the result. The error in percentage for the first sample plot was found to be 11.84%. Similarly, for the second sample plot, the error was computed to be 54.86%, which is the maximum error within all the sample plots. The main reason behind this large error is that the second plot was heavily affected by wind and rain, causing the plant to die. Since the model was generated using the condition that any other environment circumstance has not affected it, this led to a difference between actual ground scenario and the model equations, resulting in high error. Moreover, for the third sample plot, the error was found to be 10.62%. The fourth and fifth sample plots contained a lesser amount of error compared to other sample plots as they were the least affected by environmental circumstances, and the error was found to be 4.26% and 1.07%, respectively.

#### 3.6.10. Estimation of Yield from Biomass

The regression model developed between Yield and Biomass for plot was determined using a regression model, which was used to estimate the yield as tabulated below in Table 11.

The yield of each sample plot was calculated using a regression equation. In total, 70% of the data was used to construct the equation, and the rest of the data was employed to validate the results. The percentage error for the first plot was determined to be 17.51%. On the other hand, the maximum error among all the plots was found to be 62.10% for the second plot, which was due to the environmental conditions such as wind and rain that killed the plants. The error for the third plot was 16.63%. The errors for the fourth and fifth plots were found to be 7.25% and 2.47%, respectively, since these plots were least affected by environmental conditions.

#### 3.6.11. Estimation of Yield from Normalized Difference Vegetation Index (NDVI)

The regression model developed between Yield and Normalized Difference Vegetation Index (NDVI) for the plot was determined using the regression model, which was used to estimate the yield as tabulated below in Table 12.

The yield of each sample plot was estimated using a regression equation. The accuracy of the results was validated using the remaining 30% of the data. The error for the first sample plot was 20.45%; for the second sample plot, it was 58.52% due to the adverse weather conditions; for the third sample plot, it was 19.49%, and the error was 7.94% and 8.68%, respectively, for the fourth and fifth sample plot as they were less affected by the outside factors.

#### 3.6.12. Analysis of Error to Select the Parameters

Based on the observation of data directly from the field and comparing it with the yield generated from the regression model, an error for Sample Plot 4 and Sample Plot 5 was generated with various parameters to see how these parameters actually work. An error was generated by differentiating the yield generated from the field to the yield from the regression model (Table 13).

The error was visualized with the help of a graph to see the pattern of error with several parameters (Figure 14).

The error in Sample Plot 4 with several parameters such as Biomass, Plant Height, Leaf Area Index, Green–Red Vegetation Index and Normalized Difference Vegetation Index was generated. The maximum error was seen on the Normalized Difference Vegetation Index parameter, which was 9.23%, and the minimum error was seen on the Green–Red Vegetation Index parameter, which was 4.26%. This actually demonstrates that GRVI is the most important parameter in yield calculation. NDVI was derived from a Sentinel-based product. NDVI has a lower resolution compared to the other data here. This might be the reason why NDVI has deviated more from the actual value in comparison to the other parameters.

For Sample Plot 5, the same parameters were used to check the error.

Various parameters were used to predict the yield in Sample Plot 5, and the error in percentage is visualized in Table 14. The actual visualization of the error is shown in Figure 15.

The error was visualized with the help of a graph to see the pattern of error with several parameters.

## 4. Discussion

The error in Sample Plot 5 with several parameters such as Biomass, Plant Height, Leaf Area Index, Green–Red Vegetation Index and Normalized Difference Vegetation Index was generated in Figure 15. The maximum error was seen on the Normalized Difference Vegetation Index parameter, which was 8.67%, and the minimum error was seen on the Green–Red Vegetation Index parameter, which was 1.07%. This actually demonstrates that GRVI is the most important parameter in yield calculation. NDVI was derived from a Sentinel-based product. NDVI has a lower resolution compared to the other data here. This might be the reason why NDVI has deviated more from the actual value in comparison to the other parameters.

Figure 15 shows the values of Normalized Difference Vegetation Index and Green Red Vegetation Index on four different dates (25 May 2021, 25 June 2021, 15 July 2021 and 15 August 2021). NDVI is a measure of the amount of vegetation present in an area and is used to measure the health of vegetation. GRVI is a measure of the relative greenness of an area and is used to estimate the photosynthetic activity of an area.

The data shows that the NDVI values for 25 May 2021, 25 June 2021, 15 July 2021 and 15 August 2021 were 0.33, 0.75, 0.77 and 0.74, respectively. The GRVI values for these dates were 0.12, 0.46, 0.52 and 0.49, respectively. The data suggest that there was an increase in the NDVI value from 25 May 2021 to 25 June 2021, and a slight decrease from 25 June 2021 to 15 July 2021; however, the NDVI value was still higher than the initial value on 25 May 2021. There was a further decrease in the NDVI value from 15 July 2021 to 15 August 2021. The GRVI also showed an increasing trend from 25 May 2021 to 25 June 2021; however, it was followed by a slight decrease from 25 June 2021 to 15 July 2021 and a further decrease from 15 July 2021 to 15 August 2021. Overall, the data suggest that the amount of vegetation present in the area increased from 25 May 2021 to 25 June 2021 and then decreased from 25 June 2021 to 15 August 2021. The photosynthetic activity of the area also increased from 25 May 2021 to 25 June 2021 and then decreased from 25 June 2021 to 15 August 2021. The data can be used to determine the health of the vegetation in the area and to measure the photosynthetic activity of the area (Figure 16).

From the regression equation developed above, we can find several reasons why the NDVI and GRVI (Sentinel-based) values for estimating yield became deviated from the actual yield on the ground. Some of the reasons are discussed here. The first and foremost reason is that the image that was acquired with remote sensing technology, a Sentinel product, did not exactly match the date of flight of the UAV images. This can bring some changes during the development phase of the maize plant. Another reason could be the cloud coverage of the image, where reflectance is very high for the cloudy pixel. Similarly, this might also be due to the presence of mixed pixels of the satellite imagery, meaning that the plot where the maize plant was grown completely did not fall completely on one pixel but rather on multiple pixels mixed with other types. Therefore, while computing NDVI and GRVI, multiple mixed pixel effects caused NDVI and GRVI values to become abnormal. Moreover, the Sentinel image has a resolution of 10 m, which has been compared with the GRVI resolution of 0.5 cm and with the 1 m (sample plot area) yield data. The gap between satellite-based products and ground-based products is generally 10-fold, which also may be the reason behind the deviation of yield from the satellite-based NDVI and the GRVI value.

### Multiple Linear Regression Analysis

After the simple linear regression analysis, several parameters were checked to see the dependency of one on another. Sentinel-based NDVI was even plotted with Sentinel-based GRVI to see the difference. The image with same resolution (Sentinel NDVI and GRVI) did not vary much, so those two products, NDVI and GRVI (satellite-based), were not used for multiple linear regression analysis. The rest of the other parameters such as Biomass (X1), Green–Red Vegetation Index (X2), Plant Height (X3), and Leaf Area Index (X4) were only used to see the relation with the Yield (Ŷ). Here, in the equation, Yield is the dependent variable, whereas Biomass, Green–Red Vegetation Index, Plant Height and Leaf Area Index are the independent variables whose relation is shown below:Ŷ = 0.85+ 1.16X1 + 1.18X2+ 0.94X3 + 0.98X4.(3)

This equation is a multiple linear regression model that can be used to predict the Yield of a maize plant, Ŷ. The equation takes four independent variables: Biomass (X1), Green–Red Vegetation Index (X2), Plant Height (X3) and Leaf Area Index (X4) [44,45,46]. The equation assigns each of these variables a coefficient—1.16, 1.18, 0.94, and 0.98, respectively. This coefficient indicates the importance of each variable in predicting the output, with a higher coefficient indicating a greater importance [47,48]. The equation also includes a constant of 0.85, which is added to the sum of all the other terms. This constant is required to ensure that the equation is correctly centered on zero.

By multiplying each of the four variables by their respective coefficients and then summing the products, this equation is able to calculate an estimate of the maize plant yield. The larger the value of the independent variables, the more the output of the equation increases. For example, if the biomass is doubled, the output of the equation increases 1.16-fold, assuming that all other variables remain constant. Similarly, if the GRVI is doubled, the output increases 1.18-fold.

The equation can be written as Ŷ = 0.85 + 1.16X1 + 1.18X2+ 0.94X3 + 0.98X4. The constant term 0.85 is the intercept, which represents the predicted yield when all four predictor variables are zero. The coefficients of the predictor variables (1.16, 1.18, 0.94 and 0.98) indicate the amount of change in the predicted yield per unit increase of the respective predictor variable.

Biomass (X1) is the total mass of a plant, including the leaves, stems, flowers, fruits, and other parts [49,50,51]. A higher biomass indicates a larger plant size and thus can be used to predict the yield of a maize plant [52]. A coefficient of 1.16 implies that for every unit increase in biomass, the predicted yield of the maize plant increases by 1.16 units.

The Green Red Vegetation Index (X2) is an indicator of the amount of green leaf area of a crop compared to the total land area [53,54,55]. A high GRVI indicates a healthy crop with a large amount of green leaf area and a higher yield. A coefficient of 1.18 implies that for every unit increase in the GRVI, the predicted yield of the maize plant increases by 1.18 units.

Plant Height (X3) is an indicator of the size and overall growth of a plant [56,57]. A taller plant can indicate a healthier crop and thus a higher yield. A coefficient of 0.94 implies that for every unit increase in the Plant Height, the predicted yield of the maize plant increases by 0.94 units.

Leaf Area Index (X4) is the total area of a plant’s leaves relative to the ground area. A higher leaf area indicates a larger plant with a higher yield. A coefficient of 0.98 implies that for every unit increase in the Leaf Area Index, the predicted yield of the maize plant increases by 0.98 units.

This equation can be used to make predictions about the yield of maize plants, given the information about the four independent variables is provided. By adjusting the values of the four variables, the equation can then be used to understand the expected yield of a maize plant given different combinations of Biomass, GRVI, Plant Height and Leaf Area Index.

## 5. Conclusions

Three distinct strategies were employed to calculate the yield of maize plants: a ground-based method, an Unmanned Aerial Vehicle, and remote sensing technology. Several factors that impact the growth of the plants were taken into account when performing this estimation. These included Plant Height, Green–Red Vegetation Index, Leaf Area Index, Crop Surface Model, Biomass and Normalized Difference Vegetation Index. The ground-based method was used to measure the Leaf Area Index and Wet Biomass of maize plants. Photogrammetry-based methods were applied to measure the Green–Red Vegetation Index, Plant Height, and Crop Surface Model. Lastly, to assess the yield, a remote sensing-based method was used, which involved the Normalized Difference Vegetation Index and Green–Red Vegetation Index. Based on the yield generated by the model, the most and least convenient parameters were selected and compared to the actual yield from the field.

The ultimate result of the project, yield, is the product of the regression of various parameters. Ground-based data and data abstracted from the resultant post-processing of the UAV images along with the secondary data for NDVI, GRVI, and Sentinel products from a Google Earth Engine were the data sources of the project. Plant Height, GRVI, LAI, CSM, Biomass and NDVI were the used parameters for the Yield estimation. Sample yield from the field was probed with the listed parameters for the estimation of the Yield of the whole plot. The regression model, Yield vs. GRVI, has the higher regression coefficient, while Yield vs. NDVI has the lowest, as GRVI is the primary data with higher resolution, while NDVI has a low resolution of 10 m as it is abstracted from the satellite.

Finally, the conducted research estimated the yield of the maize plant. In addition, this research even helped to determine the most important parameters for estimating maize yield. The study found that the Green–Red Vegetation Index (GRVI) was the most important parameter, whereas the least important parameter was satellite-based Normalized Difference Vegetation Index (NDVI). The study also found that satellite-based NDVI was less important than UAV-based GRVI due to its lower spatial resolution and cloud cover. The study concluded that GRVI is the most important parameter for estimating maize yield and that future research should focus on incorporating higher-resolution NDVI data, genomic information, management practices, and environmental data into yield estimation models.

Here is a summary of the key points:GRVI is the most important parameter for estimating maize yield.Satellite-based NDVI is less important than UAV-based GRVI due to its lower spatial resolution and cloud cover.Future research should focus on incorporating higher-resolution NDVI data, genomic information, management practices, and environmental data into yield estimation models.

## Figures and Tables

**Figure 1 sensors-23-05432-f001:**
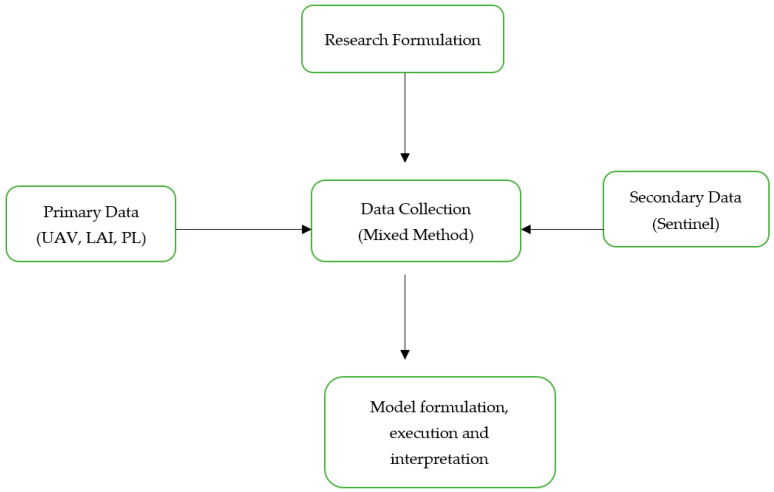
Conceptual Research Design Framework.

**Figure 2 sensors-23-05432-f002:**
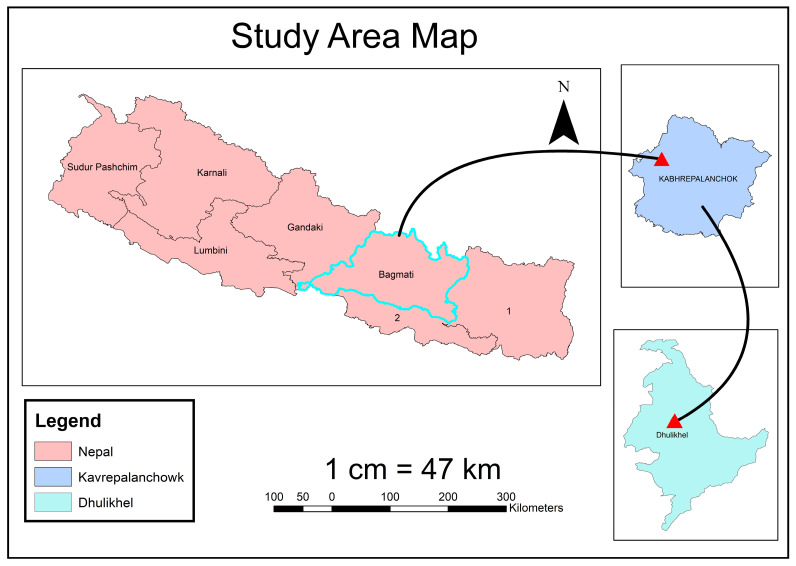
Study Area Map (Data Source: Survey Department, Nepal).

**Figure 3 sensors-23-05432-f003:**
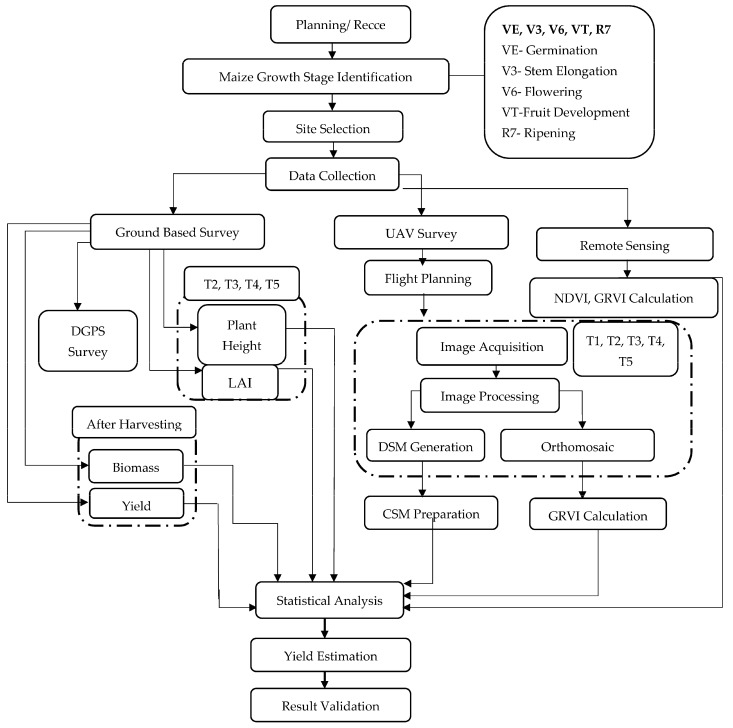
Overall Methodology.

**Figure 4 sensors-23-05432-f004:**
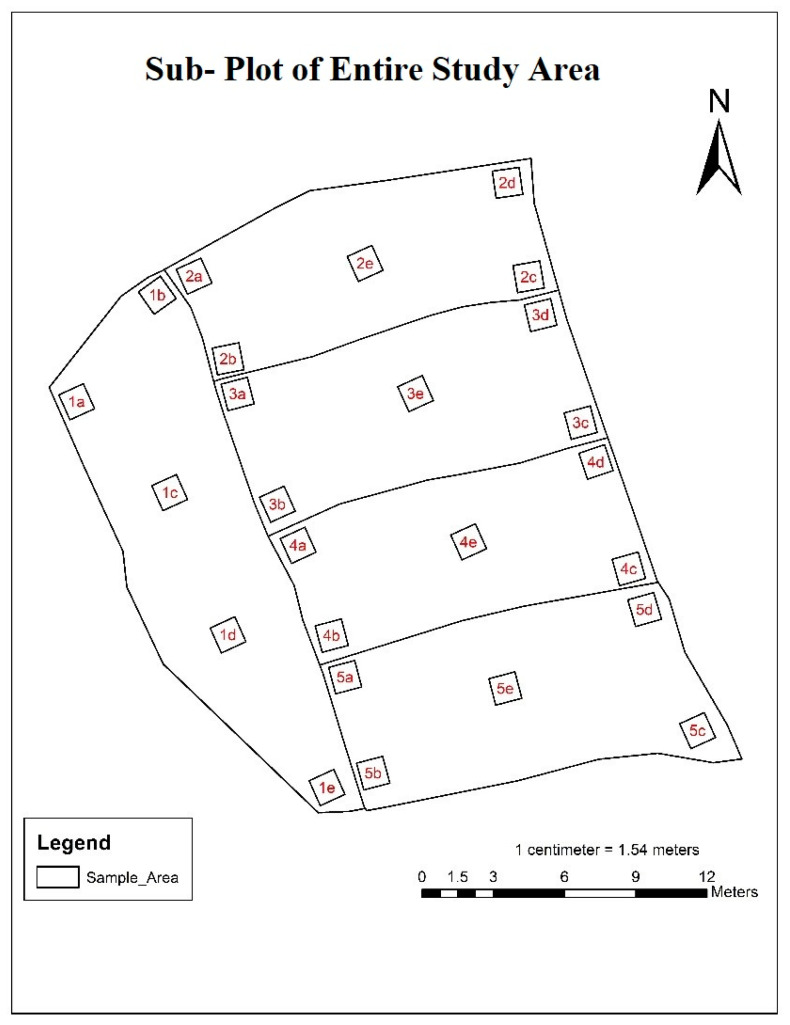
Sample Area of Entire Study Area.

**Figure 5 sensors-23-05432-f005:**
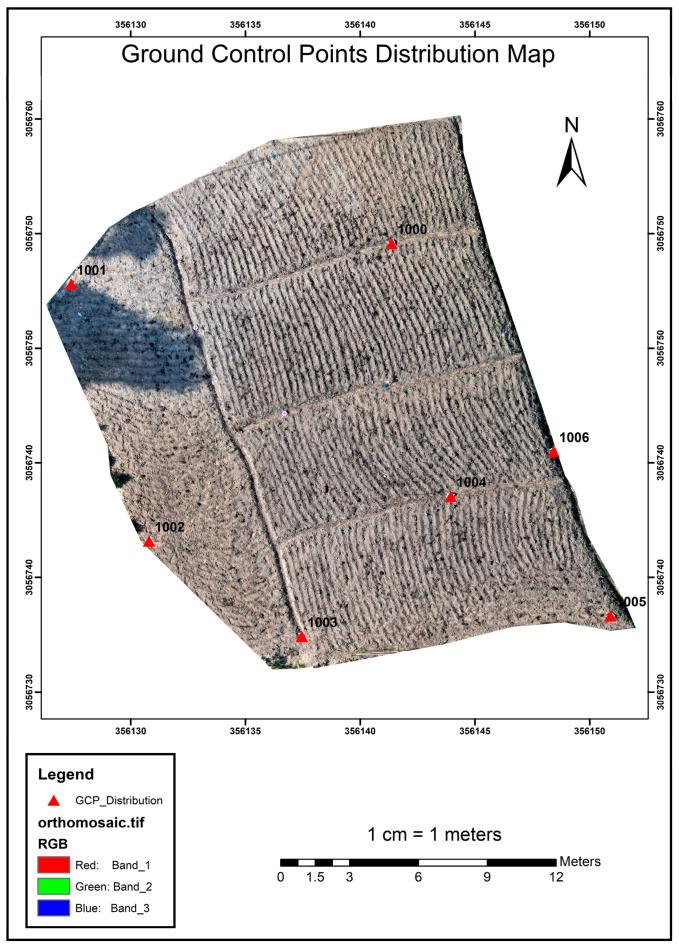
Ground Control Point Distribution Map.

**Figure 6 sensors-23-05432-f006:**
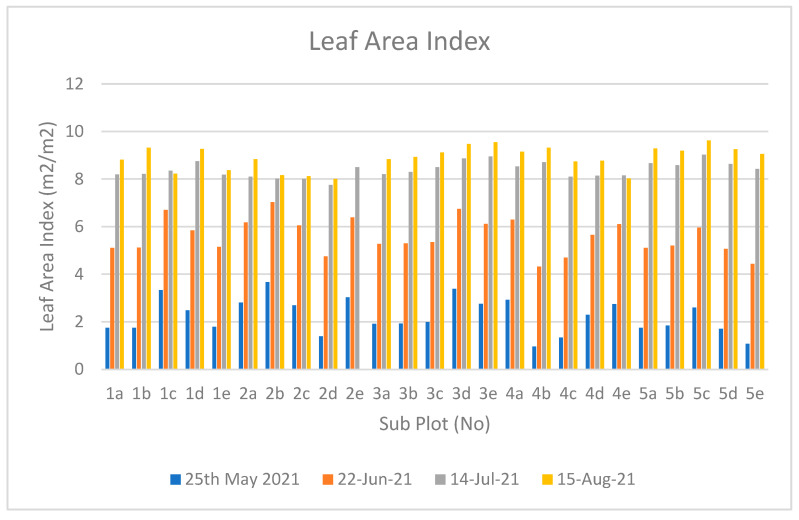
Leaf Area Index of Plot in Different Growth Stages.

**Figure 7 sensors-23-05432-f007:**
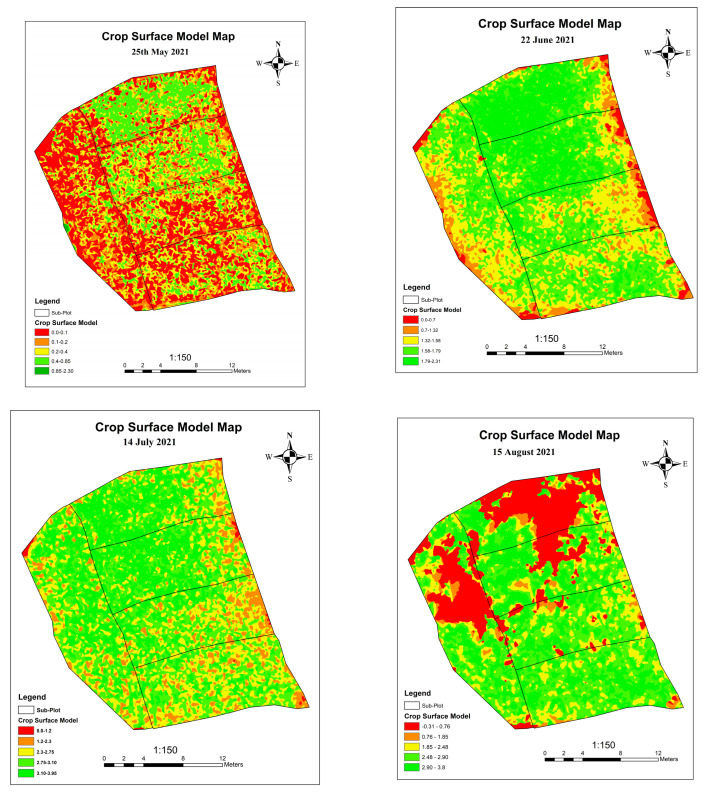
CSM in 2nd, 3rd, 4th and 5th Phase of cycle.

**Figure 8 sensors-23-05432-f008:**
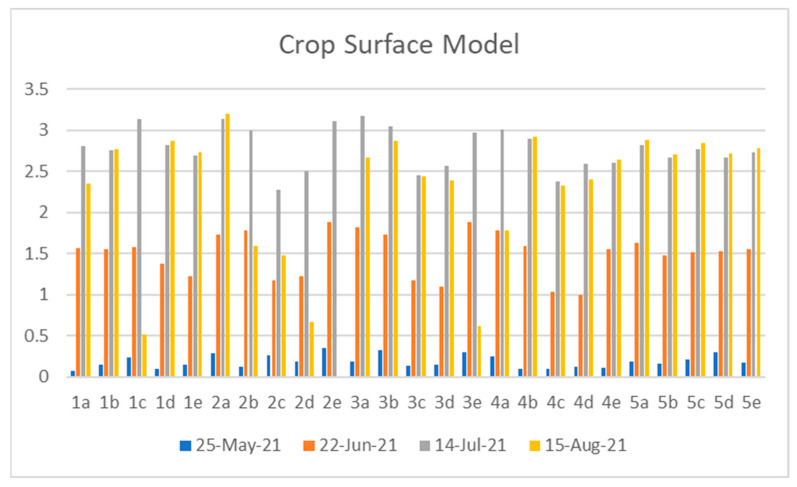
Plant Height at Different Stages.

**Figure 9 sensors-23-05432-f009:**
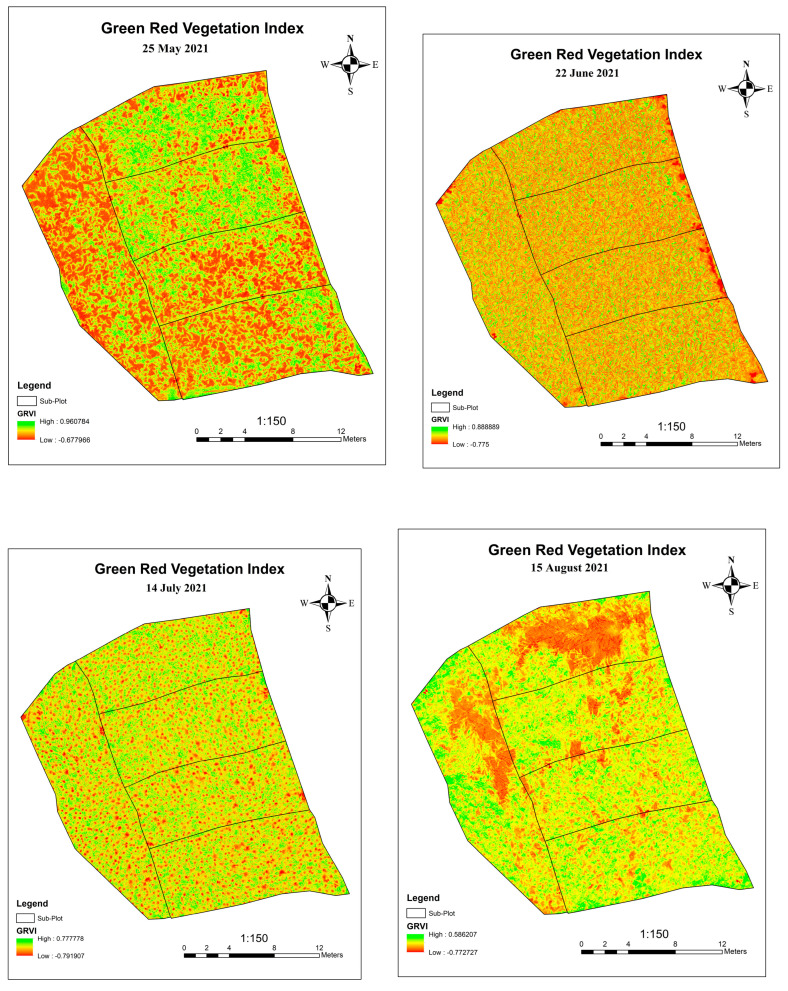
GRVI Map of 2nd, 3rd, 4th and 5th Time Frame.

**Figure 10 sensors-23-05432-f010:**
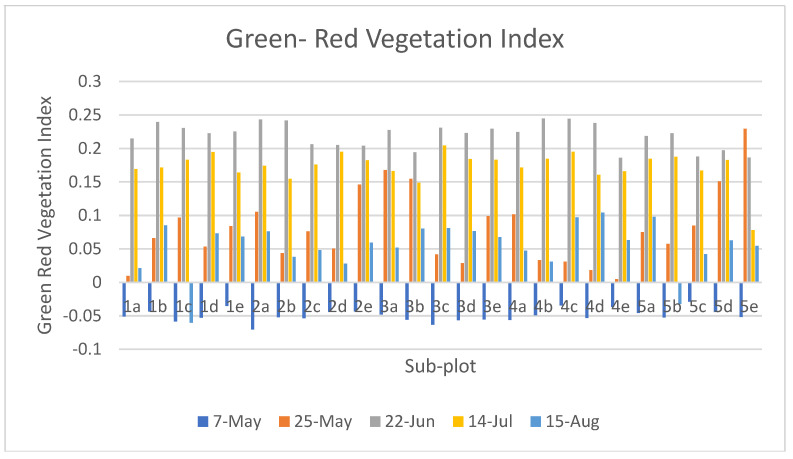
Green–Red vegetation Index of Plot at Different Growth Stages.

**Figure 11 sensors-23-05432-f011:**
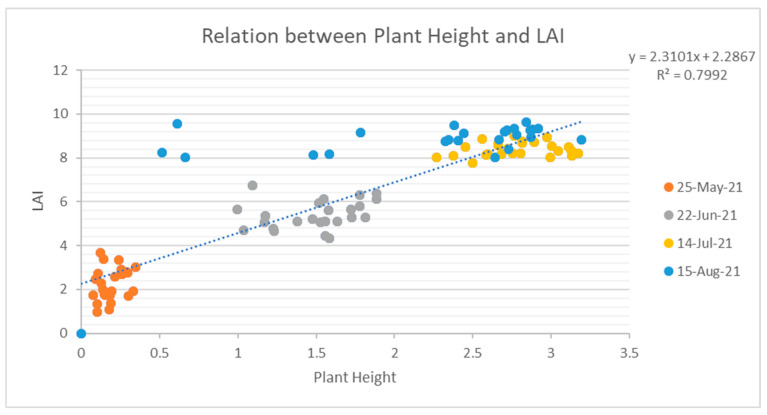
Graph of Plot Showing Relation Between Plant Height and Leaf Area Index.

**Figure 12 sensors-23-05432-f012:**
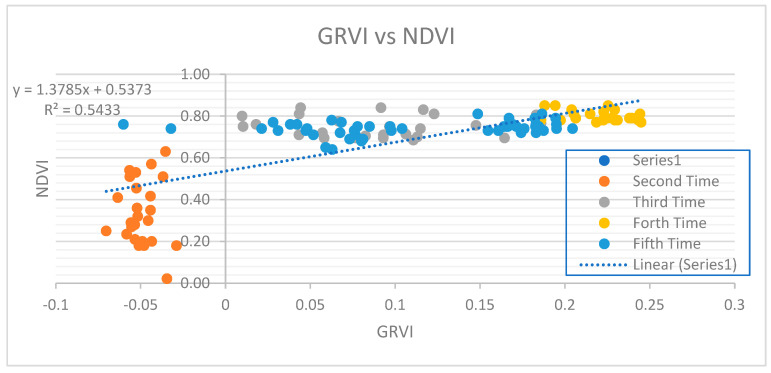
Graph of Plot Showing Relation Between GRVI and Leaf NDVI.

**Figure 13 sensors-23-05432-f013:**
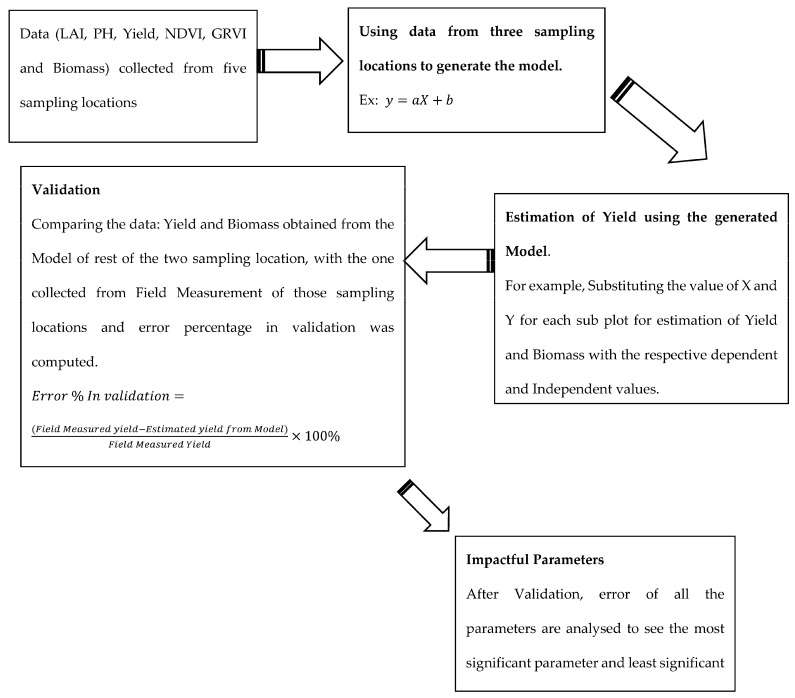
Model Generation Methodology.

**Figure 14 sensors-23-05432-f014:**
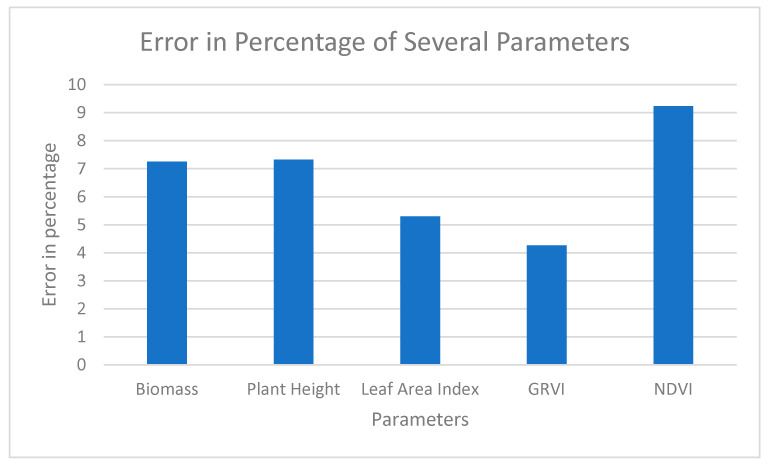
Error Distribution from Several Parameters for Sample Plot 4.

**Figure 15 sensors-23-05432-f015:**
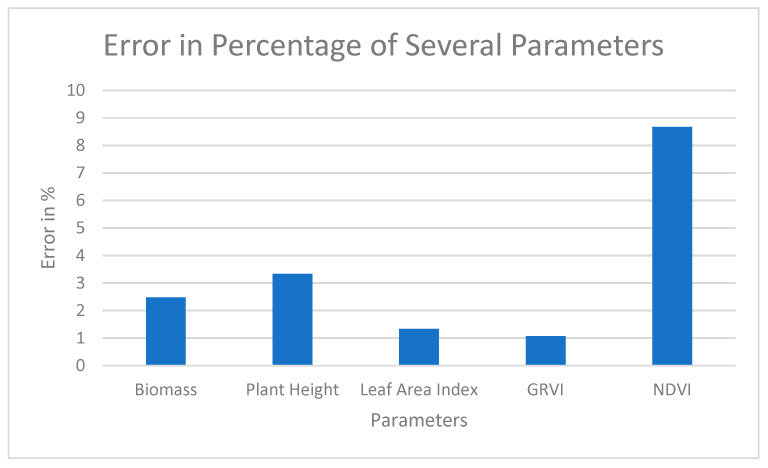
Error Distribution from Several Parameters for Sample Plot 5.

**Figure 16 sensors-23-05432-f016:**
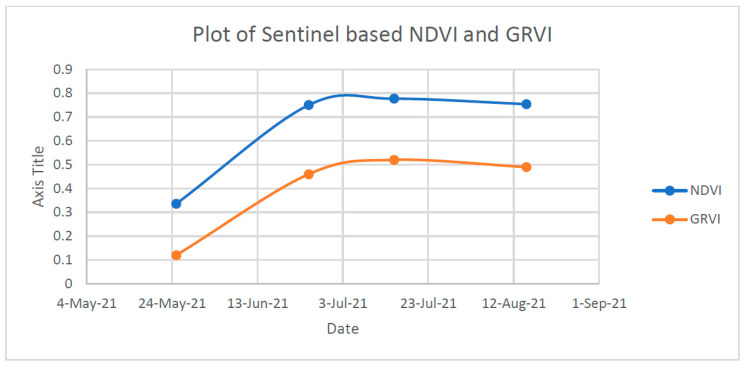
NDVI vs. GRVI (Satellite-Based Product).

**Table 1 sensors-23-05432-t001:** Coordinate values of GCPs of Study Area.

Station	Easting (m)	Northing (m)	Elevation (m)
1000	357,478.466	3,058,765.321	1384.572
1001	357,486.091	3,058,774.183	1384.052
1002	357,475.308	3,058,776.268	1384.392
1003	357,483.792	3,058,764.365	1384.442
1004	357,472.468	3,058,763.249	1384.663
1005	357,281.086	3,058,753.376	1383.392
1006	357,471.174	3,058,755.588	1383.414

**Table 2 sensors-23-05432-t002:** Regression Model for Estimating Yield using Plant Height.

Yield vs. PH		
Plot	Regression Model	R^2^
1	y = 0.6701x + 5.3548	0.99
2	y = 0.3541x + 2.2827	0.98
3	y = 0.0489x + 3.6412	0.87
4	y = 0.4462x + 8.0399	0.78
5	y = 0.8608x + 2.4781	0.91

**Table 3 sensors-23-05432-t003:** Regression Model for Estimating Yield Using LAI.

**Yield vs. LAI**		
**Plot**	**Regression Model**	**R^2^**
1	y = 0.9118x − 3.986	0.97
2	y = 0.5615x − 1.3953	0.97
3	y = 0.1286x + 2.4286	0.96
4	y = 0.9941x − 4.6506	0.79
5	y = 0.2603x + 2.5337	0.98

**Table 4 sensors-23-05432-t004:** Regression Model for Estimating Yield using GRVI.

Yield vs. GRVI		
Plot	Regression Model	R^2^
1	y = 0.1749x − 0.4314	0.86
2	y = 0.0401x + 0.3185	0.97
3	y = 0.4382x + 1.7112	0.90
4	y = 0.0104x + 0.1358	0.95
5	y = 0.4617x + 2.4024	0.99

**Table 5 sensors-23-05432-t005:** Regression Model for Estimating Yield using Biomass.

Yield vs. Biomass		
Plot	Regression Model	R^2^
1	y = 1.3239x − 8.285	0.61
2	y = 1.4286x − 7.4748	0.99
3	y = 1.121x − 6.3167	0.88
4	y = 1.2521x + 3.2791	0.99
5	y = 1.2949x + 2.8891	0.98

**Table 6 sensors-23-05432-t006:** Regression Model for Estimating Yield using NDVI.

Yield vs. NDVI		
Plot	Regression Model	R^2^
1	y = 2.0345x + 0.6871	0.68
2	y = 2.1429x + 1.2471	0.69
3	y = 2.0172x − 6.2	0.54
4	y = 2.0348x + 0.6658	0.63
5	y = 2.5967x − 2.0623	0.69

**Table 7 sensors-23-05432-t007:** Regression Model for Estimating Yield using satellite-based GRVI.

Yield vs. GRVI		
Plot	Regression Model	R^2^
1	y = 1.3210x + 0.8372	0.58
2	y = 1.3211x + 2.3219	0.59
3	y = 1.9211x − 4.9321	0.56
4	y = 1.9821x + 0.3921	0.50
5	y = 1.7328x − 3.3911	0.62

**Table 8 sensors-23-05432-t008:** Validation of Yield using Plant Height in every sample plot.

Yield vs. Plant Height (PH)			
Yield from Model	Yield from Field	Error	Error in Percentage
415.21	340.86	0.2181	21.81
281.33	169.19	0.6627	66.27
310.63	259.62	0.1964	19.64
268.21	289.40	0.0732	7.32
410.98	397.71	0.0333	3.33

**Table 9 sensors-23-05432-t009:** Validation of yield using Leaf Area Index in every sample plot.

Yield vs. LAI			
Yield from Model	Yield from Field	Error	Error in Percentage
389.0922	340.86	0.1415	14.15
268.1471	169.1932	0.5848	58.49
300.1429	259.6268	0.1560	15.6
274.0614	289.406	0.0530	5.30
402.0112	397.7108	0.0108	1.08

**Table 10 sensors-23-05432-t010:** Validation of yield using Green–Red Vegetation Index in every sample plot.

Yield vs. GRVI			
Yield from Model	Yield from Field	Error	Error in Percentage
381.2192	340.86	0.1184	11.84
262.0212	169.1932	0.5486	54.86
287.2132	259.6268	0.1062	10.62
277.0614	289.406	0.0426	4.26
401.9821	397.7108	0.0107	1.07

**Table 11 sensors-23-05432-t011:** Validation of yield using Biomass in every sample plot.

Yield vs. Biomass			
Yield from Model (kg)	Yield from Field (kg)	Error	Error in Percentage
400.5723	340.86	0.1751	17.51
274.2775	169.1932	0.6210	62.10
302.8247	259.6268	0.1663	16.63
268.4224	289.406	0.0725	7.25
407.5621	397.7108	0.0247	2.47

**Table 12 sensors-23-05432-t012:** Validation of yield using NDVI in every sample plot.

Yield vs. NDVI			
Yield from Model (kg)	Yield from Field (kg)	Error	Error in Percentage (%)
410.5723	340.86	0.2045	20.45
268.2130	169.1932	0.5852	58.52
310.2510	259.6268	0.1949	19.49
312.3870	289.406	0.0794	7.94
432.2130	397.7108	0.0867	8.67

**Table 13 sensors-23-05432-t013:** Error in Sample Plot 4 with Various Parameters.

Parameters	Sample Plot	Yield from Field (kg)	Yield from Model (kg)	Error in Percentage
Biomass			268.422	7.25
Plant Height			268.219	7.32
Leaf Area Index	4	289.406	274.061	5.30
GRVI			277.061	4.26
NDVI			312.387	9.23

**Table 14 sensors-23-05432-t014:** Error in Sample Plot 5 with Various Parameters.

Parameters	Sample Plot	Yield from Field (kg)	Yield from Model (kg)	Error in Percentage (%)
Biomass	5		407.562	2.47
Plant Height			410.982	3.33
Leaf Area Index		397.712	402.011	1.08
GRVI			401.982	1.07
NDVI			432.213	8.67

## Data Availability

The data supporting the findings of this study are available from the first author upon reasonable request.

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
