# Peer review of "Growth Monitoring and Yield Estimation of Maize Plant Using Unmanned Aerial Vehicle (UAV) in a Hilly Region"

_sensors, 2023, doi:10.3390/s23125432_

Round 1

Reviewer 1 Report

Weather conditions such as light and wind speed at the time of data collection with the UAV are not provided in the paper. The weather conditions at the time of collection should be kept similar for the four phases.

Only linear analysis was used, is non-linear modeling more consistent with the growth of maize.

Inconsistent descriptions of growth stages and measurement times.

Yield and biomass for the final period can be calculated by post-harvest, but how are actual yield and biomass calculated for the intermediate period.

NDVI and GRVI (line 22), DTM, LAI (line 26), VE, V3, V6, T1, T2, T3 (line 146-170 in the figure) are unspecified in their first appearance.

Line 200-202. How leaf area is measured.

The formula for LAI GRVI etc. is not stated.

Line 467-478, 487-498, 511-522 are similar in description.

Line 203, 206, 219 etc. The units of variables in the graph are missing.

Line 206-208. The basis for the faster growth of site 5 is not evident in the graph without data to illustrate.

Line 210. Please provide the specific number of days of growth of maize in the four periods and the basis for this division.

Line 246, 250. Why is June 22 the second stage and May 25 the third stage ?

Author Response

This is the response to the Reviewer 1

Reviewer 2 Report

This manuscript used UAV images to monitor growth and then estimate yield of maize. Overall, the used method in this study is reasonable and this study suites the scope of journal of sensors. Finally, yield estimation results have high accuracy. There are some major concerns that should be addressed before acception.

(1) all figures and tables should have a numbered name, and each should be mentioned and explained in the main text.

(2) It is better if figures of crop surface model at different times have the same legend. GRVI figures should also have the same legend. In addition, the equation of GRVI should be presented.

(3) To improve readability, regression Models for Estimating Yield using Plant Height, LAI, GRVI, and Biomass should be shown in one table.

(4) Regression Models developed from satellite-derived NDVI and GRVI should be shown in one table.

(5) Sentinel-2 satellite data were used? More information on Sentinel data should be described in Materials and Methods section.

(6) It seems that satellite-based yield estimations show lower accuracy than UAV-based model. This may due to the spatial mismatch between satellite data (10-20m) and ground plot (1m). Thus, it is interesting if this issue is discussed.

Author Response

This is the reply to reviewer 2

Reviewer 3 Report

Recommendations to authors

General comments:

(1) Similar scientific works exist in literature, so authors must explain the reason for conducting this study emphasizing the novelty of current work because now I think it is not clear enough.

(2) Authors must follow the instructions given by the Journal (online). Several changes must be applied throughout the text. 

(3) Tables and figures seem to be hastily constructed, not following the corresponding Journal instructions.

(4) Statistical significance information for the independent variables in the yield equation is missing. Please provide further statistical data to support the estimation equation for yield. 

(5) English must be improved in several parts.

Detailed comments:

- Line 2: Maybe you should consider adding “RGVI” to the title.

- Line 26: Improve the keywords; try to avoid abbreviations and words that exist already in the title.

- Lines 69 – 73: I believe that the aim of this work is not to present crop variability (that affects crop production) but to make a comparison between the two indices NDVI and GRVI, if I am not mistaken. Therefore, I expect authors at the end of introduction to explain the reason for conducting this study emphasizing the novelty of current work.

- Line 85: You must provide the equation for the calculation of GRVI values (although it is known). For example: GRVI = (Green − Red) / (Green + Red). The same applies for NDVI (NDVI = (NIR − Red) / (NIR + Red)).

- Line 118: Label format for figures seems not appropriate/compatible with the template of the Journal. All figures must be numbered and there must be a reference in the text for each one presented. Please check the instructions given by the Journal online. The same applies to all figures.

- Line 188: Something goes wrong with the font size of the heading.

- Line 197: Table format needs to be improved to follow the standards set by the Journal. Please the instructions given by the Journal online. The same applies to all tables.

- Line 197: “Stations” or “Ground Control Points (GCPs)”? Why are these “stations” not plotted on the map above that shows the entire area? Please add them and maybe consider adding their coordinates in index and remove the table you now have.

- Line 203: Remove figure outline and improve the elements of the graph (set font color to black, change font and follow all the instructions given by the Journal).

- Line 227: Something goes wrong with the paragraph indentation.

- Line 303: Obviously this is not "Figure 1". Please check all figures and make corrections. The graph seems to be hastily constructed. Please improve their quality. Check and follow the instructions given by the Journal online. 

- Line 196, 357, 364: Tables must be numbered too. There must be a reference in the text for each table presented. Please check all tables and the corresponding paragraphs throughout the text. All tables must follow the instructions given by the Journal. Please check and make corrections.

- Line 197: Change the format of table according to the instructions given the Journal online.

- Line 364: R2 values must have 3 decimals. The same applies to all tables presenting R2 values.

- Line 414: Two or three decimals maximum in values is enough.

- Line 525: Why only for plots 4 and 5? What about the rest?

- Line 535: Check the font used in the table.

- Line 539: Apply sorting to the graph.

- Line 554: If there are duplicate values for all entries (sample plot, yield from field) then it means they should be removed, Add this information to the caption of the table.

- Line 562: Apply sorting to the graph.

- Line 574: The graph seems to be hastily constructed. Please improve their quality. Dates for example cannot be read.

- Line 622: Are the independent parameters statistically significant? (If any of them are not statistically significant, then they must be removed from the equation). Please provide more statistical information to support the estimate using the equation you provide.

- Line 622: Please add (Eq.1) to the right side.

- Lines 664 – 669: These are not conclusions and must be removed. 

- Line 664: Improve the English of the paragraph. There is no point in using references in the conclusions section. This is a place strictly for your conclusions based on your results and discussion.

Author Response

This is the reply to reviewer 3

Round 2

Reviewer 1 Report

Thank the authors for accepting my suggestions or comments. However, as a scientific paper, there are still several issues needing revisions before publication.

1. There are as more as 22 figures and 14 tables in the paper. I am wondering if they are all necessary, or is it appropriate to merge some figures to one, such as Figures 7-10 (not just here), they are the crop surface models at different time frames. Showing the figures in the present way is not readable.

2. Still for the figures. The authors added the units for some figures, but there are still some figures missing units (e.g., figures 16-18). I strongly suggest the authros to go through the whole text and check the figure and/or table one by one.

3. Some formatting issues, such as the different font sizes in the tables (e.g., table 7 and 8); X1, X2... the number should be subscript.

4. One suggestion. In the response to the reviewers, the authors should point out the specific position of their revisions, with the page number and line number. Or it is difficult to find them though the authors highlighted them in the manuscrip.

Author Response

Reply to the reviewer 1 (round 2)

Reviewer 2 Report

I don't have further comments on this manuscript.

Author Response

No comment was there to reply.

Reviewer 3 Report

Recommendations to authors

General comments:

(1) Similar scientific works exist in literature, so authors must emphasize the novelty of the manuscript because now I think it is not clear enough.

(2) Authors must follow the instructions given by the Journal (online). Several changes must be applied throughout the text. Although detailed recommendations were given, authors did not follow the instructions given by the Journal.

(3) Regarding Tables and figures: authors improved some parts but both Tables and figures still do not follow the corresponding Journal instructions.

(4) Statistical significance information for the independent variables in the yield equation is missing. Authors must provide further statistical data to support the estimation equation for yield. This was mentioned at the previous stage, but no information was added.

(5) English must be improved in several parts.

Detailed comments:

- Keywords: avoid abbreviations and words that exist already in the title.

- Comment from the previous stage of review: I believe that the aim of this work is not to present crop variability (that affects crop production) but to make a comparison between the two indices NDVI and GRVI, if I am not mistaken. Therefore, I expect authors at the end of introduction to explain the reason for conducting this study emphasizing the novelty of current work.

- Regarding the labels of the figures: caption format of figures was improved but it is still not appropriate/compatible with the template of the Journal. All figures must be numbered and there must be a reference in the text for each one presented. Remove expressions like “figure below” and give the exact number of the figure. Please check the instructions given by the Journal online. The same applies to all figures.

- In the results section the font size of all sub-titles is wrong. Please make the corrections needed based on template of the Journal.

- Regarding the format of the tables: the format has been improved but several unneeded spaces or wrong line spacing still exist and the format of the tables needs to be further improved to follow the standards set by the Journal. Please the instructions given by the Journal online. The same applies to all tables.

- Figure 5: Regarding “Stations” or “Ground Control Points (GCPs)” you must add their coordinates in index or add a grid with coordinates. In its current form, no one can say where this area is located.

- Something goes wrong with the indentation of several paragraphs.

- Comment from the previous stage of review: Tables must be numbered too. There must be a reference in the text for each table presented. Please check all tables and the corresponding paragraphs throughout the text. All tables must follow the instructions given by the Journal. Please check and make corrections.

- Check the way equations are presented; please add (Eq.1) to the right side of the sentence instead of “(Equation 1).

- Improve the English of the last paragraph in conclusions. 

Author Response

Response to reviewer 3 (2nd round)
